# Turbulence Induced Cloud Voids: Observation and Interpretation

Katarzyna Karpinska[1], Jonathan F.E. Bodenschatz[2], Szymon P. Malinowski[1], Jakub L. Nowak[1], Steffen Risius[2], Tina Schmeissner[4], Raymond A. Shaw[3], Holger Siebert[4], Hengdong Xi[2], Haitao Xu[2], and Eberhard Bodenschatz[2]

[1]Institute of Geophysics, Faculty of Physics, University of Warsaw, Warsaw, Poland
[2]Max Planck Institute for Dynamics and Self-Organization, Goetingen, Germany
[3]Michigan Technological University, Houghton, Michigan, USA
[4]Leibniz Institute for Tropospheric Research, Leipzig, Germany

**Correspondence:** S. P. Malinowski (malina@fuw.edu.pl)

**Abstract.** The phenomenon of "cloud voids", i.e., elongated volumes inside a cloud that are devoid of droplets, was observed with laser sheet photography in clouds at a mountain-top station. Two experimental cases, similar in turbulence conditions yet with diverse droplet size distributions and cloud void prevalence, are reported. A theoretical explanation is proposed based on the study of heavy inertial sedimenting particles inside a Burgers vortex. A general conclusion regarding void appearance is drawn from theoretical analysis. Numerical simulations of polydisperse droplet motion with realistic vortex parameters and Mie scattering visual effects accounted for can explain the presence of voids with sizes similar to that of the observed ones. Clustering and segregation effects in a vortex tube are discussed for reasonable cloud conditions.

## 1 Introduction

The dynamics of heavy inertial particles in turbulent flow is a universal problem that appears in astrophysics, oceanography, engineering and atmospheric sciences. In particular, in cloud physics, deeper understanding of the interaction between atmospheric turbulence and cloud droplets is seen as a potential answer to many important questions (Bodenschatz et al. (2010)). Over the years, there has been considerable speculation about the possible role of coherent, long-lived vortex structures in cloud turbulence and microphysical processes, including both condensational growth and collision-coalescence growth (see Tennekes and Woods (1973), Maxey and Corrsin (1986), Shaw et al. (1998), Shaw (2000), Markowicz et al. (2000), Hill (2005)). This paper describes the first in situ observation of a phenomenon we refer to as "cloud voids" - cylindrical volumes devoid of droplets recorded in real clouds - and is focused on determining whether inertially induced voids indeed may occur in clouds due to the presence of strong vortex structures.

Vortex tubes, sometimes called "worms", are severely intermittent, coherent, elongated and long-lasting structures characteristic of high Reynolds number turbulent flows (e.g. Mouri et al. (2000)). Past theoretical and experimental studies lack general

conclusions about their characteristic time and length scales, intensity and appearance in turbulence. In particular, most of the research was conducted under conditions different from multiscale atmospheric turbulence. Statistical analysis based on such research (see e.g. Jiménez et al. (1993), Belin et al. (1999), Pirozzoli (2012)) showed that Burgers vortex core size $\delta$ (defined later in Eq. 2), scales roughly with the Kolmogorov length scale $\eta$:

$\delta = m\eta$                                                                        (1)

and that $m$ has a distribution ranging from a few to a few tens with its mean around $m = 4$. Moisy and Jimenez (2004) analyzing DNS instant velocity fields propose that vorticity structures' geometrical aspect ratios evolve towards long tubes (1:1:10) with increasing vorticity threshold. What is more they show that these structures concentrate into clusters of the size in inertial range of scales. This implies the presence of large-scale organization of the small-scale intermittent structures. In the study by

Biferale et al. (2000), statistics of vortex filament lifetime for a low Taylor microscale Reynolds number $Re_\lambda$ indicate that the maximum lifetime is on the order of the integral timescale, whereas its mean scales with the Kolmogorov timescale. Numerical experiments described in Tanahashi et al. (2008) suggest that there is a relation between root mean square velocity fluctuations and the circulation parameter $\Gamma$ of a vortex tube modeled as a Burgers vortex.

Previous efforts to study dynamics of heavy, inertial particles in vortices were made by simulating droplet trajectories in a prescribed velocity field for several simple single vortex models. Such research for the simplest model of a line vortex with stretching was conducted by Markowicz et al. (2000) with limitation to horizontally oriented vortices. In order to better understand the problem of cloud droplet dynamics in atmospheric conditions the same model but with arbitrary gravity alignment was studied by Karpinska and Malinowski (2014). Another model, free from the problem of unrealistic singularity on the

vortex axis, is a Burgers vortex with stretching. It is commonly seen as a very good approximation of a real vortex tube (Neu (1984), Jimenez and Wray (1998)). The specific features of droplet motion for monodisperse droplets in a Burgers vortex were examined by Hill (2005) for horizontal alignment and by Marcu et al. (1995) for arbitrary alignment with respect to gravity. Clustering and segregation effects are of importance when talking about interaction between particles and turbulent flow. We define clustering of particles of arbitrary sizes as an inhomogeneous distribution of particles in space. Segregation refers to the

cases in which the spatial distributions of different sized particles are anticorrelated. Different kinds of particle clustering in turbulent flow were reviewed by Gustavsson and Mehlig (2016), turbulence-induced segregation was for example treated by Calzavarini et al. (2008).

In this paper we describe the serendipitous observation of numerous, isolated voids in clouds, while conducting measurements at a mountain-top station. The voids were visually striking and generated great excitement from the scientific team. Here, we

present direct, 2D observations of the distinct types of patterns of clear air in clouds, along with accompanying turbulence and cloud microphysical measurements. We pose the question of whether the observed cloud voids are consistent with inertial droplet response to turbulence under atmospheric conditions. To answer this question, analysis of the particle dynamics in a Burgers vortex is further developed for the heavy sedimenting polydisperse case and the model is interpreted in the context of

the observations.

The paper is structured as follows. Section 2 describes the measurement method and analysis of experimental data that shows the phenomenon of cloud voids. Section 3 presents relevant features of single droplet trajectories in a vortex tube model. Section 4 extends the analysis to the motion of polydisperse droplets in a vortex to define the conditions of cloud void emergence. Section 5 outlines the influence of Mie scattering theory on droplet imaging and the implications for void observation. Section 6 presents void numerical simulation results. Section 7 incorporates the discussion and conclusions.

## 2  Experimental Evidence

Intriguing structures inside clouds, as presented in Fig. 1, were recorded by means of laser sheet photography during observations performed on 27 and 29 August 2011 at Umweltforschungsstation Schneefernerhaus (UFS) on the slopes of Mt. Zugspitze in the German Alps. Each time, the cloud event lasted for several hours. For a description of the observatory and characterization of the cloud and turbulence conditions on-site, see Risius et al. (2015) and Siebert et al. (2015). Authors of these papers showed that turbulence and cloud microphysical properties at the measurement site are quite reasonable representations of measurements made in 'free' clouds away from the surface.

Clouds were illuminated by a laser sheet created with a frequency-doubled high-power Nd:YAG laser (532 nm, 45 W). The sheet was set either vertical or oblique with respect to gravity. The angle between the laser sheet plane and camera recording plane in the oblique case was chosen to increase the scattering intensity on droplets and falls within the range of 30-40 degrees. The laser sheet in the observation region was around 50 cm wide and 1 cm thick. Images covering the approximately 2 m long section of the sheet at a distance approximately 10 m from the source were taken with a Nikon D3S DSLR 12 Mpix camera.

Laser sheet photography was accompanied by high-resolution measurements of small-scale turbulence and cloud microphysics, as described in Siebert et al. (2015). Flow turbulence was measured by 3D ultrasonic anemometers operated at 10 Hz, from which the velocity structure functions were calculated using Taylor's frozen-flow hypothesis, and the mean energy dissipation rates were determined using inertial range scaling. Droplet size was measured by a PDI probe (Chuang et al., 2008) mounted approximately 6 m down from the camera level. Figure 2 presents the measurement set-up and recorded size spectra of droplets. Droplet and turbulence measurements are summarized in Table 1. The mean values refer to 30-minute long record corresponding to the camera acquisition series.

Two kinds of events in which droplet spatial distribution is visibly inhomogeneous were distinguished in the collected images. The first kind was characterized by an irregular interface separating clear-air and cloudy-air volumes and/or cloudy volumes of visibly different properties over a wide range of spatial scales (panel b) in Fig. 1). Inhomogeneities of the second kind, present within the cloudy volumes, were called cloud voids in "Swiss cheese" clouds. Cloud voids were small (a few centimeters scale), the interface was usually blurry (see panels a) and c) in Fig. 1) and the shapes of clear-air regions were often

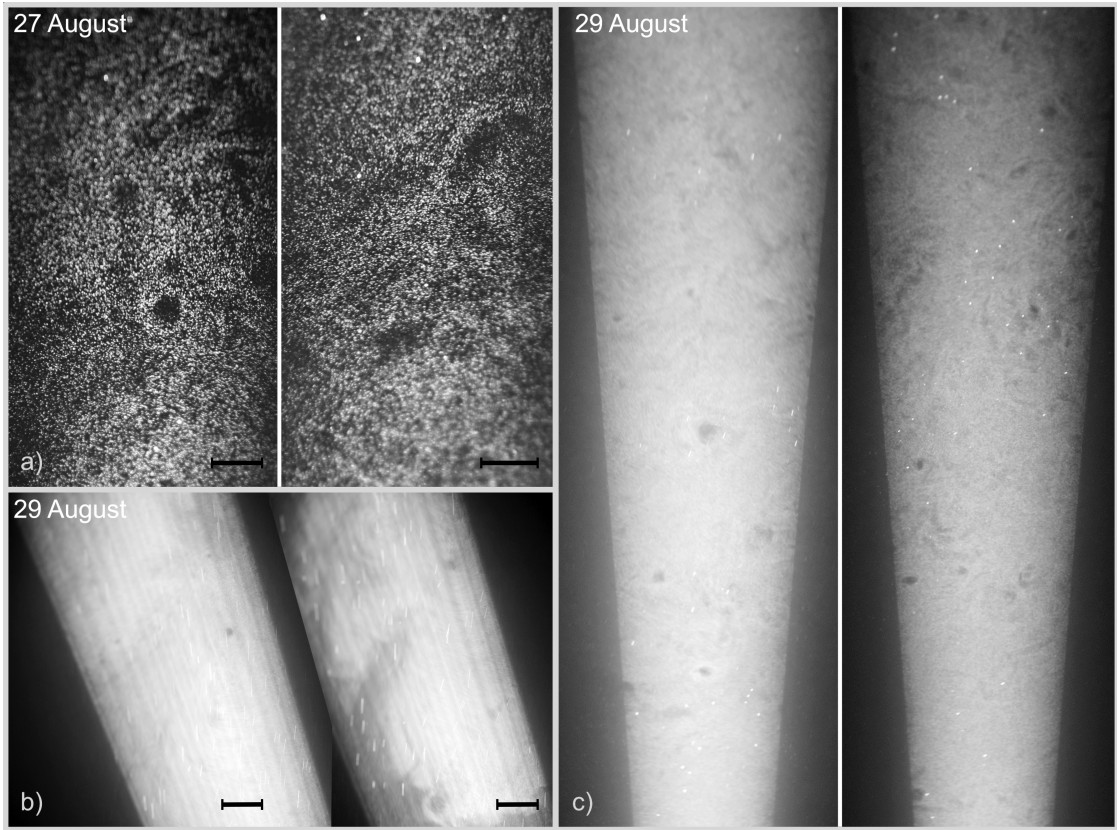

**Figure 1.** Examples of cloud voids observed at the UFS station with various camera-laser configurations. Images taken on 27 August (panel a) were chosen to estimate cloud void sizes. The ones recorded on 29 August evening (panel b) show the difference between inhomogeneities produced by cloud voids and those resulting from the mixing with clear air at the cloud edge. Other images from 29 August (panel c) suggest that the voids can be quite frequent in the sample volume. Bright spots and lines are due to presence of larger precipitation particles. 10 cm long segment is shown to represent spatial scale assumed in the void size calculation. For more details, see the movies attached in the supplementary materials.

close to round or elliptic (see magnified voids in Fig. 3). It is important to point out that the more intuitive expression "cloud holes" with regards to these inhomogeneities is avoided on purpose because it is commonly used referring to the cloud-free regions occurring in stratocumulus decks, as described for example in Gerber et al. (2005). Inhomogeneities of the first kind are argued to be created in the process of cloud – clear-air mixing (e.g. Warhaft (2000)). In contrast, in some series of images and movies, the shape of the recorded tracks of cloud droplets suggest the following cloud void origin hypothesis: they result from interactions between inertial, heavy cloud droplets and small-scale vortices present in a turbulent cloud. Comparison of the two described cases becomes straightforward when conducted on the basis of the movies in the database (Karpinska et al., 2018). In the movie "ms01" between 13 s and 22 s there are two cloud void appearances. Motion of the void in the homogeneous

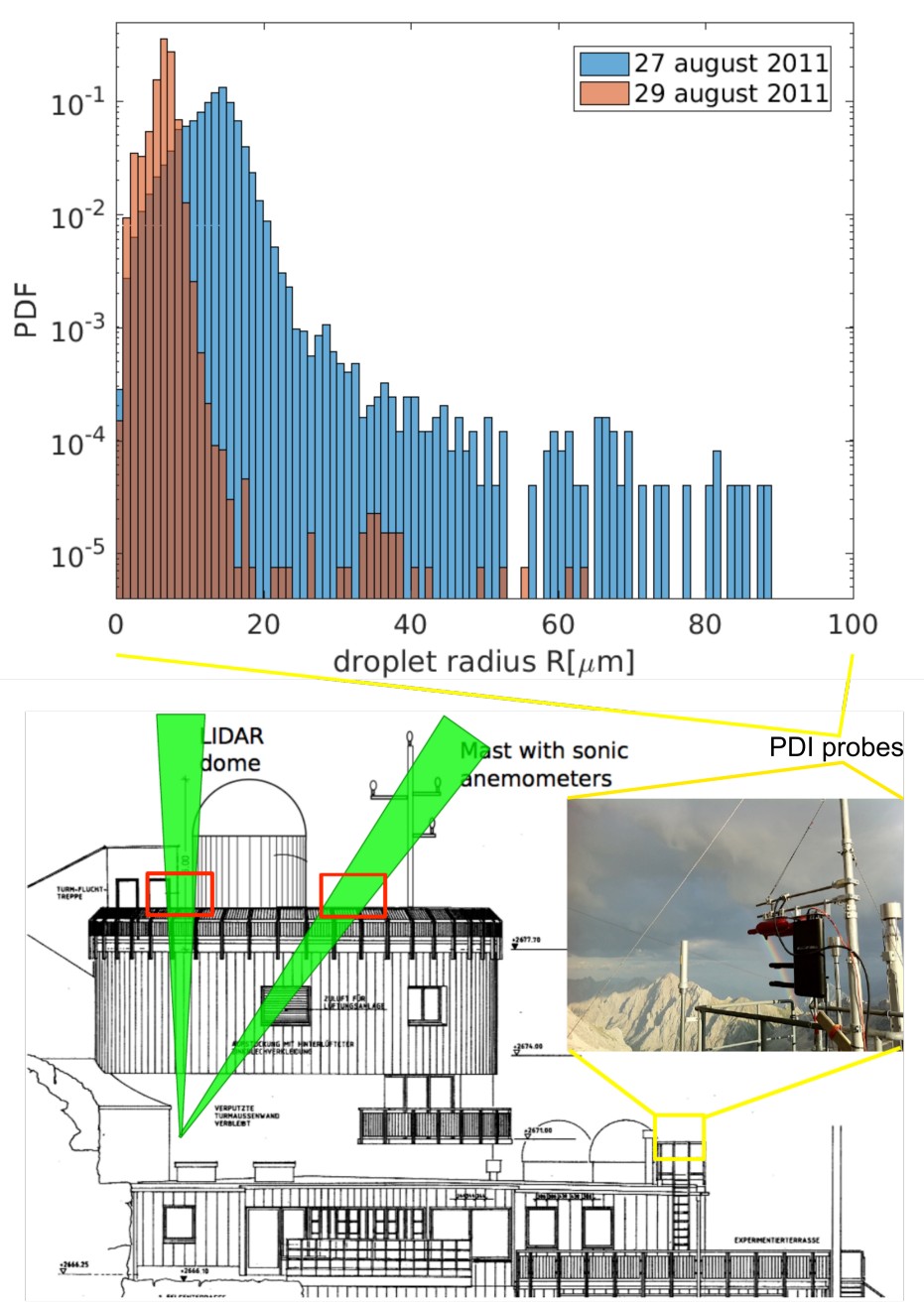

**Figure 2.** Droplet size distributions measured with a PDI probe (top) and the arrangement of instruments at the measurement site (bottom).

cloud field resembles motion of a worm. Movie "ms02" presents cloudy and clear air mixing at the cloud edges.

**Table 1.** Properties of turbulence and cloud droplets during observation periods.

|  | August 27th | August 29th |
|---|---|---|
| Energy dissipation rate $\epsilon$ [m$^2$/s$^3$] | 0.055 | 0.070 |
| Kolmogorov length scale $\eta$ [mm] | 0.50 | 0.47 |
| Komogorov timescale $\tau_\eta$ [s] | 0.017 | 0.015 |
| Mean droplet radius $R$ [$\mu$m] | $12.9 \pm 4.8$ | $6.4 \pm 1.5$ |
| Sauter mean radius $R_{32}$ [$\mu$m] | 18.1 | 7.3 |
| Stokes number $St$ (mean) | 0.126 | 0.035 |
| Stokes number $St_{32}$ (Sauter) | 0.247 | 0.045 |
| Sedimentation parameter $Sv$ | 0.676 | 0.172 |
| Froude number $Fr$ | 0.431 | 0.470 |
| Number density $n$ [cm$^{-3}$] | $56 \pm 47$ | no data |

There were a few series of cloud void images collected with various laser-camera settings on the two experimental days. The best quality series, made in the morning of the 27th, was chosen for void size analysis. For the series of 17 photos selected for analysis, there were four in which voids were not clear enough to be accounted for. In the remaining 13 photos 27 voids were identified. Each one's size was manually determined. In the case of a round void, the diameter was taken as the

5 size; in a case of flattened or ellipsoidal void, the maximal chord was taken. The typical void diameter was estimated to be 3.5±1 cm; the maximal, 12±4 cm; the minimal, 1±0.5 cm. Images from the analyzed series from the morning of August 27th showing examples of objects identified as voids are presented in the panel a) of Fig. 1. Voids captured on the 29th of August were not analyzed due to the large uncertainty resulting from the unknown geometry of the camera-laser set-up. The general experimental observation was that the voids were smaller then those on August 27th. Definitive experimental verification of

10 the cloud void origin is not possible on the basis of collected data only; however, in next sections, we argue that void creation due to inertia of droplets present inside vortex tubes is highly probable.

## 3    Motion of Heavy, Inertial Particles in the Vortex Tube Model

To address theoretically the issue of cloud void origin, the concept of the polydisperse inertial droplet population response to a coherent vortex pattern is followed. Here the first step on this track is taken by presenting relevant features of single droplet

trajectory in the chosen vortex tube model. A population of particles is assumed to form a dilute collection of material points, heavy and inertial, displaced by gravity force and viscous force (Stokes drag) only. Burgers vortex (Burgers, 1948) is used as a model of a steady vortex tube. The $z$-axis in the cylindrical coordinate system $(r, \varphi, z)$ is aligned with the vortex axis. 3D velocity field $\boldsymbol{v}$ is determined by two parameters: circulation $\Gamma$ and stretching strength $\gamma$:

$$\boldsymbol{v} = -\frac{\gamma}{2} r \hat{e}_r + \frac{\Gamma}{2\pi r} \left( 1 - e^{-\frac{r^2}{2\delta^2}} \right) \hat{e}_\varphi + \gamma z \hat{e}_z, \tag{2}$$

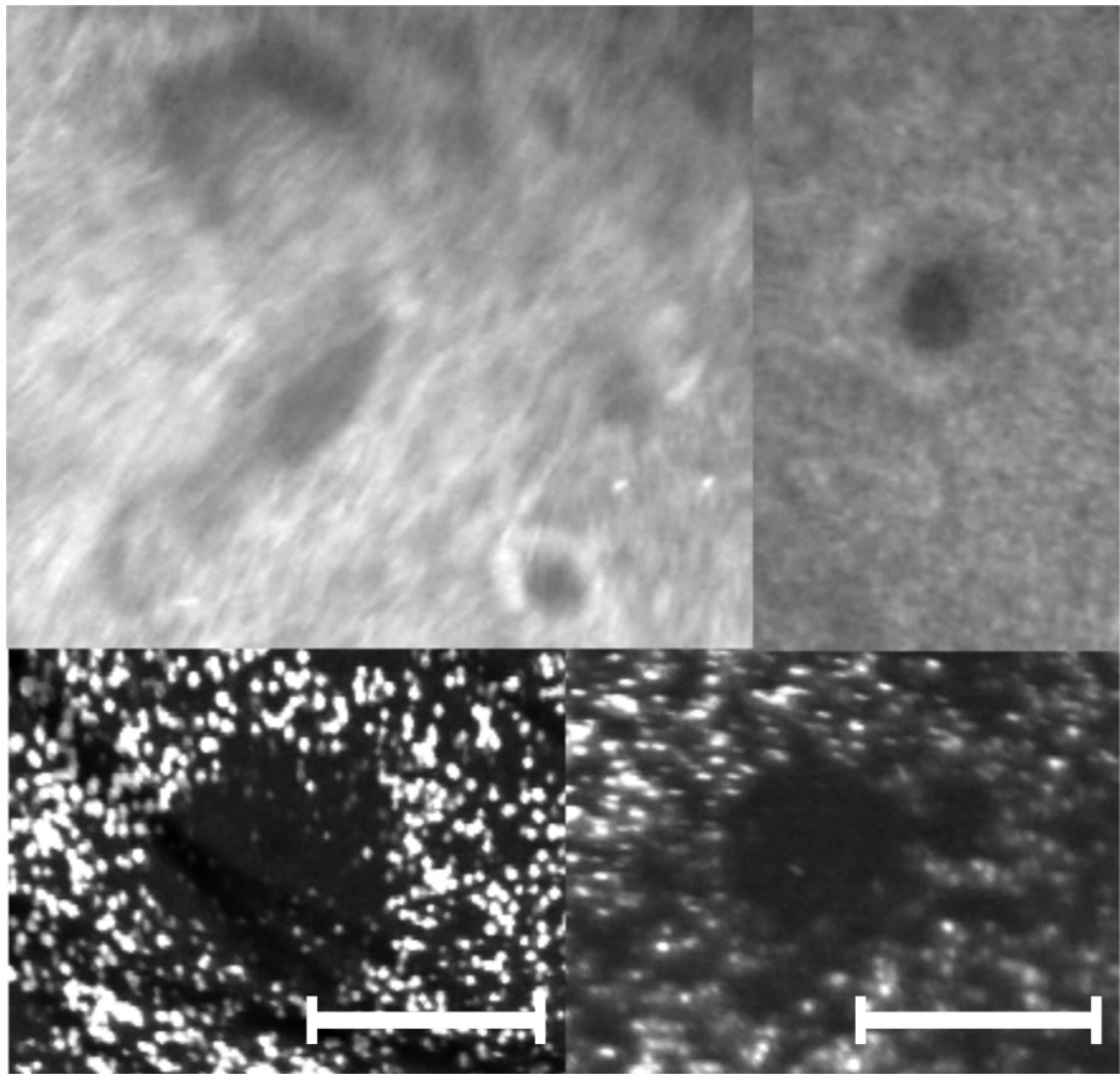

**Figure 3.** Example close-ups of variously shaped cloud voids observed at the UFS station with different camera-laser configurations. 5 cm long segment is shown to represent spatial scale assumed in the void size calculation.

where $\delta = \sqrt{2\nu/\gamma}$ is the vortex core size and $\nu$ denotes the kinematic viscosity. A particle's equation of motion is as follows:

$$\ddot{\boldsymbol{r}} = \frac{1}{\tau_p} \left( \boldsymbol{v} - \dot{\boldsymbol{r}} \right) + \boldsymbol{g}, \tag{3}$$

where $\tau_p$ is the particle relaxation time and $\boldsymbol{g} = -g \left( \sin\theta \hat{e}_y + \cos\theta \hat{e}_z \right)$ is gravitational acceleration inclined by the angle $\theta \in [0, 90^o]$ with respect to the vortex axis.

5   The analysis of single droplet motion in projection on a plane $(r, \varphi)$ perpendicular to the vortex axis (henceforth called 2D

space) was conducted by Marcu et al. (1995). Here it is summarized and completed with in-depth evaluation.

The behavior of a droplet inside the vortex depends on a set of six parameters $\{\Gamma, \gamma, \theta, \tau_p, g, \nu\}$. The nondimensionalization of the equations leads to Eq. 4 and gives a set of dimensionless parameters $\{St, Sv, \theta, A\}$:

$$
\begin{cases}
\ddot{r}^* - r^*\dot{\varphi}^{*2} = -St^{-1}\left(Ar^* + \dot{r}^* + Sv\sin\varphi\right) \\
2\dot{r}^*\dot{\varphi}^* + r^*\ddot{\varphi}^* = St^{-1}\left(\frac{1}{2\pi r^*}(1 - e^{-\frac{r^{*2}}{2}}) - r^*\dot{\varphi}^* - Sv\cos\varphi\right) \\
\ddot{z}^* = St^{-1}\left(Az - \dot{z}^* - Sv\cot\theta\right)
\end{cases} \tag{4}
$$

Henceforth, dimensionless variables are denoted by $^*$. Stokes number $St$ here is calculated with the use of the characteristic timescale of the Burgers vortex flow, which is the vortex core rotation time $\tau_f = \delta^2\Gamma^{-1}$, so $St = \tau_p\tau_f^{-1} = \tau_p\Gamma\delta^{-2}$. The sedimentation parameter $Sv$ is a ratio of $\tau_f$ to the timescale of sedimentation through the vortex: $Sv = \tau_f\tau_g^{-1} = \delta g\tau_p\sin\theta\Gamma^{-1}$. It characterizes motion in a plane perpendicular to the vortex axis. The last quantity $A = \nu\Gamma^{-1} = 1/Re_v$ is the nondimensional strain parameter, the inverse of vortex Reynolds number $Re_v$. It is worth mentioning that the ratio of Stokes number to the sed-

imentation parameter, called Froude number $Fr = StSv^{-1}$, is a measure of the influence of gravitational force on the droplet motion. In the limit of a large Froude number, gravity is considered negligible.

As one can see in Eq.4, the equation describing particle motion along the vortex axis is independent from the equations describing motion in 2D space. Thus these components can be analysed separately.

Motion along the vortex axis is determined by stretching outflow drag and gravity force. As a consequence, the particle

$z$ position shows an exponential dependence on time. What is more, every droplet has one unstable equilibrium point at $z_0^* = SvA^{-1}\cot\theta = \nu^{-1}g\delta\tau_p cos\theta$. The analytical solution was presented in (Karpinska and Malinowski, 2014). The solutions of Eq.4 in 2D space have several different attractors. It is helpful to distinguish two cases: with gravity and without gravity (valid as well when gravity is parallel to the vortex axis).

For the case of a vertical vortex or no gravity, for every particle of a given radius a stable, circular periodic orbit exists if

$St < St_{cr} = 16\pi^2 A$. For $St \geq St_{cr}$, there exists a stable equilibrium point positioned on the vortex axis. A radius of the periodic orbit satisfies the equation:

$$
r^{*2}\sqrt{A/St} - \left[1 - \exp\left(-r^{*2}/2\right)\right]/2\pi = 0. \tag{5}
$$

Motion in 2D space with gravity included is much more complicated. Nonparallel alignment of the gravity vector and vortex axis ($\theta \neq 0$) destroys the axial symmetry of the system and introduces the presence of other attractors, such as limit cycles and

equilibrium points outside the axis. The rest of this section is devoted to thorough analysis of these attractors.

**Table 2.** Existence of equilibrium points with respect to $A$ and $Sv$ parameters. $A_c r$, $r_s^*$, $r_{min}^*$, $Sv_{min}$, $Sv_{max}$ are defined in the text body.

| $A \geq A_{cr}$ | $Sv$ - arbitrary | 1 eq. point |
|---|---|---|
| | $< Sv_{min}$ | 1 eq. point at $r^* < r_s$ |
| $A < A_{cr}$ | $[Sv_{min}, Sv_{max}]$ | 2 or 3 eq. points |
| | $> Sv_{max}$ | 1 eq. point at $r^* > r_{min}$ |

**Table 3.** Burgers vortex nondimensional numbers

| | |
|---|---|
| $A_{cr}$ | 0.02176 |
| $r_i^*$ | 2.1866 |
| $r_s^*$ | 1.585201 |
| $A_t$ | 0.01917 |

For a nonzero $\theta$, every particle always has equilibrium points in 2D space. Positions of these points are determined by $Sv$ and $A$. They can be obtained by solving the equation for the radial component $r^*$:

$$r^* A \sqrt{1 + \left( \frac{1 - \exp\left(\frac{-r^{*2}}{2}\right)}{2\pi A r^{*2}} \right)^2} = Sv. \tag{6}$$

Now, let $f_A(r^*)$ be the left hand side of Eq. 6. A plot of this function for a given $A$ is called an equilibrium curve (see Fig.2 in
Marcu et al. (1995)). Detailed analysis of equilibrium curve dependence on parameters is performed in the next paragraph and leads to the conclusions summarised in Table 2.

It is easy to find that $f_A(0) = 0$ and $\lim_{r^* \to \infty} f_A(r^*) = \infty$. Moreover, there exists a critical value $A_{cr}$ for which bifurcation from one unique solution (for $A \geq A_{cr}$) to maximally three solutions (for $A < A_{cr}$) occurs. $A_{cr}$ corresponds to the equilibrium
curve that has a horizontal slope at the inflection point. $A_{cr}$ value was estimated numerically (see the Table 3).
For $A \geq A_{cr}$ the equilibrium curve is a monotonically increasing function of $r^*$ so there exists exactly one solution for every $Sv$ value. For $A < A_{cr}$ the equilibrium curve always has one maximum at $r_{max}^*$ and one minimum at $r_{min}^*$. The inflection point at $A = A_{cr}$ on the equilibrium curve lies at $r_i^*$ (see the Table 3). It restricts values of $r_{max}^*$ from above and values of $r_{min}^*$ from below. Consequently, for $Sv < f_A(r_{min}^*)$ and for $Sv > f_A(r_{max}^*)$, there is only one solution. For $Sv_{min} = f_A(r_{min}^*)$ and
for $Sv_{max} = f_A(r_{max}^*)$, there are two solutions. For $f_A(r_{min}^*) < Sv < f_A(r_{max}^*)$, there are three solutions.

Not only is the existence of the solutions important but their stability as well. Let $r_0$ denote an arbitrary solution of Eq. 6. The exact form of the stability condition of the solution $r_0^*$ is governed by the function $\phi(r_0^*)$ (as defined in Marcu et al.

**Table 4.** Stability conditions of particle equilibrium points present in the Burgers vortex with respect to vortex strain parameter $A$ and dimensionless distance from the vortex axis $r^*$. $A_t$, $\varphi(r^*)$, $r_s^*$, $r_{min}^*$ and $r_{max}^*$ are defined in the text body.

| | $\leq r_s^*$ | $(r_s^*, r_{max}^*)$ | $[r_{max}^*, r_{min}^*)$ | $\geq r_{min}^*$ |
|---|---|---|---|---|
| $A < A_t$ | unstable for $St > A/\|\phi(r_0^*)\|$ | stable | partly unstable | stable |
| $A \geq A_t$ | | stable | | |

(1995)). The condition can take two different forms depending on the sign of this function

$$\phi(r_0^*) = \frac{1}{(2\pi)^2} \left[ \frac{1 - \exp(-r_0^{*2}/2)}{r_0^{*2}} \right] \left[ \frac{1 - \exp(-r_0^{*2}/2)}{r_0^{*2}} - \exp(-r_0^{*2}/2) \right]. \tag{7}$$

The function has only one zero at $r_s^*$ (see Table 3). For small radii ($r_0^* < r_s^*$), the equilibrium is stable if:

$$\frac{St}{A} \leq \frac{1}{|\phi(r_0^*)|}. \tag{8}$$

For greater radii ($r_0^* > r_s^*$), the condition for stability depends explicitly only on $A$:

$$A \geq \sqrt{\phi(r_0^*)}. \tag{9}$$

The equilibrium point satisfying the first type of the condition was shown to be a focus, the second type to be a node. Analysis of the equilibrium point stability conditions by Marcu et al. (1995) is expanded in the next paragraph with emphasis on the dependence on strain parameter $A$. The results are summarised in Table 4.

Generally, stability conditions are different for $A$ ranges separated by $A_t = \max_{r_0^*}\left(\sqrt{\varphi(r_0^*)}\right)$ (approximate value obtained numerically in Table 3). $A > A_t$ always satisfies then the condition expressed by 9. The term "partly unstable" in the table refers to the following: In the range of $r_{max}^* \leq r_0^* < r_{min}^*$, for a given $A$, only a small fraction of the total range (near points $r_{max}^*$ and $r_{min}^*$) is stable. This range grows with increasing $A$. Numerical experiments show, however, that their domain of attraction in the presence of other stable points (at least one exists always) is relatively small.

The combination of multiple existence conditions with stability conditions creates a variety of single particle motion scenarios. Some of them are shown in Fig.4 in Marcu et al. (1995). These scenarios are used in the Sect.4 to carry out a search for vortex model parameter values that produce a void. Fig. 4 here illustrates one of these scenarios in which there are three equilibrium points: I - unstable point near the axis, II - unstable middle distance point, III - stable point far from the axis.

A particle, depending on initial position and velocity, rotates around point I or is weakly attracted by unstable point II or is strongly attracted by stable point III.

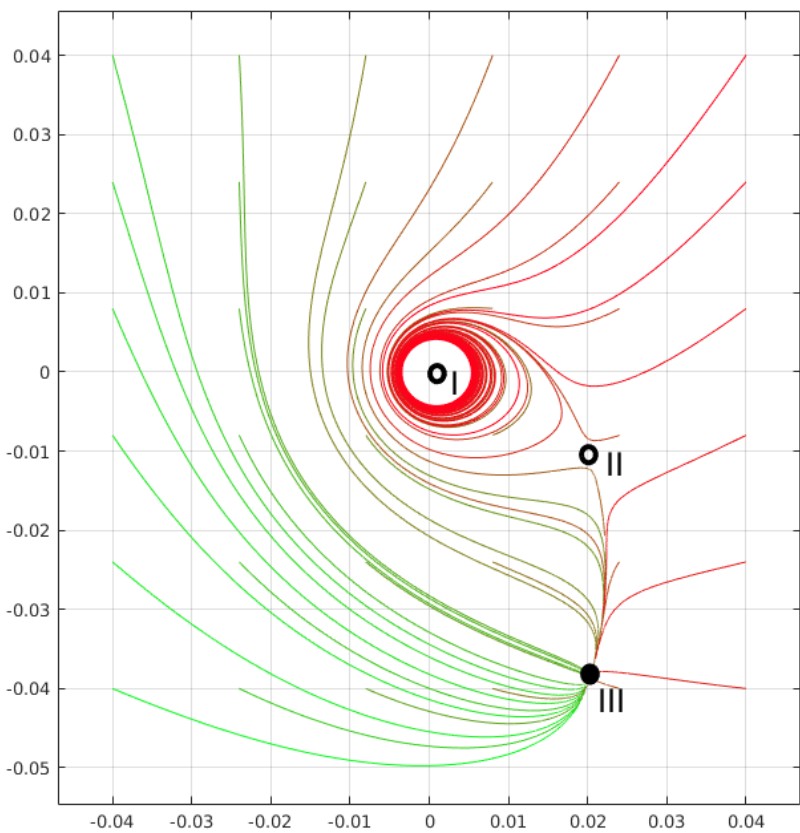

**Figure 4.** Example illustration of monodisperse droplets trajectories in 2D space (projection on a plane perpendicular to the vortex axis) with zero initial velocity. Particle trajectories (color lines) show the presence of three equilibrium points marked with black dots: I - unstable near vortex axis, II - unstable middle distance point, III - stable point far from the axis. Some droplets rotate around the vortex core on various orbits, some of these orbits may correspond to limit cycles. Different attraction regions can be noticed.

## 4 Cloud void creation conditions

Using conclusions concerning the motion of a single particle, the following hypothesis on polydisperse particle collective behaviour can be formulated: a void can be created if a majority of the droplets have an unstable equilibrium point close to the axis $r_0^* < r_s^*$, leading to a limit cycle or periodic orbit attraction. The radius of curvature should be large enough for a void to
be noticeable. If multiple equilibrium points exist, attraction by a stable equilibrium point far from the axis $r_0^* > r_{min}^*$ should not considerably influence droplet trajectories close to the void considerably. The first and the last condition are inspected in the Subsect.4.1, the second condition in the Subsect.4.2.

### 4.1 Polydisperse particles

Obtaining a mathematically strict condition for creation of an arbitrary sized void in arbitrary polydisperse collection of droplets
would be too detailed and too complicated to be profitable for the interpretation of crude experimental results. Thorough analysis of single droplet motion in addition to what was presented in the paper (Marcu et al., 1995)) was performed and used to draw approximate conclusions about polydisperse droplet motion.

The most obvious conclusion is that when circulation of the vortex is too small: $A \geq A_{cr}$, the motion of particles is determined mostly by the gravitational force and resembles sedimentation through the vortex with curved trajectories. Circulation must
then be large enough: at least $A \leq A_{cr}$ for void to be created.

The other condition is that equilibrium points near the vortex center for the majority of the particles are unstable, allowing circulation around the vortex axis. Inequality 8 is exploited here to find the qualitative dependence between vortex and particle parameters that fulfills this condition.

First, distance from the vortex axis of the equilibrium point should be in the range $r^* \leq r_s^*$. It allows approximation of relation
described by Eq. 6 in the vicinity of $r^* = 0$. The approximation is made with the assumption that in this vicinity the dependence on $A$ is weak (see Fig.2 in Marcu et al. (1995)).

$$\left(\frac{Sv}{A}\right)^2 = r^{*2}\left[1 + \left(\frac{1 - \exp\left(-r^{*2}/2\right)}{2\pi A r^{*2}}\right)^2\right] = r^{*2} + \frac{(1 - \exp\left(-r^{*2}/2\right))^2}{(2\pi A r^*)^2} \simeq r^{*2} + \frac{1}{4\pi^2 A^2}\frac{r^{*2}}{4} = r^{*2}\left(1 + (4\pi A)^{-2}\right), \quad (10)$$

so in the end:

$$r_0^* \simeq 4\pi Sv \left(1 + (4\pi A)^2\right)^{-\frac{1}{2}}. \quad (11)$$

Secondly, the function $\phi(r_0^*)$ (see Fig.3 in Marcu et al. (1995)) is approximated in the chosen $r^*$ range by linear dependence on $r^*$ : $\phi(r_0^*) \simeq -(1 - r_0^*/r_s^*) \cdot \left(16\pi^2\right)^{-1}$. At $r^* = 0$ it has the same form as obtained for the case without gravity in Marcu et al. (1995). The above approximations are used simplifying the stability condition determined by Eq. 8. In the end the condition for unstable points near the axis are algebraicly transformed and expressed by splitting it in two parts. The first part concerns only the vortex parameters and the second part the particle sizes.


The first part requires that strain parameter $A$ is small enough:

$$A < A_{max} \propto B^{1/3} \tag{12}$$

and consequently circulation $\Gamma$ large enough:

$$\Gamma > \Gamma_{min} \propto B^{-1/3} \tag{13}$$

The second part demands that particle Stokes number falls within the range $(St_1, St_1 + \Delta St)$ and it is related to the vortex parameters (results shown to the leading term order in $A$):

$$\begin{aligned} St_1 &\propto A \\ \Delta St &\propto B A^{-2} \end{aligned} \tag{14}$$

$B$ is a new dimensionless parameter depending on vortex core size $\delta$ and gravity influence $g\sin\theta$:

$$B = \underbrace{\frac{r_s^*}{2^8 \pi^3}}_{const} \frac{\nu^2}{g\sin\theta \delta^3}. \tag{15}$$

The maximal strain parameter $A_{max}$ (minimal circulation $\Gamma_{min}$) increases (decreases) weakly with $B$. So the larger the vortex core size $\delta$ and gravity influence $g\sin\theta$ the larger the minimal circulation needed.

The following conclusions can be drawn from the above approximate relations:

- There is a threshold (minimal) value of circulation needed for void creation. It increases with inclination angle ($\sin\theta$) and vortex size $\delta$.

- The greater the circulation the smaller particles have their unstable points near the axis.

- The range of particles having unstable points near the axis increases with increasing circulation and decreases with increasing gravity influence and vortex size $\delta$.

Building up on these results it may be concluded that it is harder to observe voids created by horizontally aligned vortices than vertically aligned ones and also more difficult to observe voids the larger particle size range is.

## 4.2 Particle orbit in a void - the radius of curvature

In order to obtain a void, a majority of the droplets must circle around the axis and the curvature of their trajectories should be large enough for a void to be noticeable. In order to estimate the curvature radius we perform the following reasoning. For simplicity particle and vortex constants and parameters are now chosen to match those of water droplets in the cloudy air and henceforward particles are called droplets. Firstly, a basic vortex spatial scale is established for the measurement conditions. Premises found in the literature discussed in the introduction were used for making the assumption that the proportionality

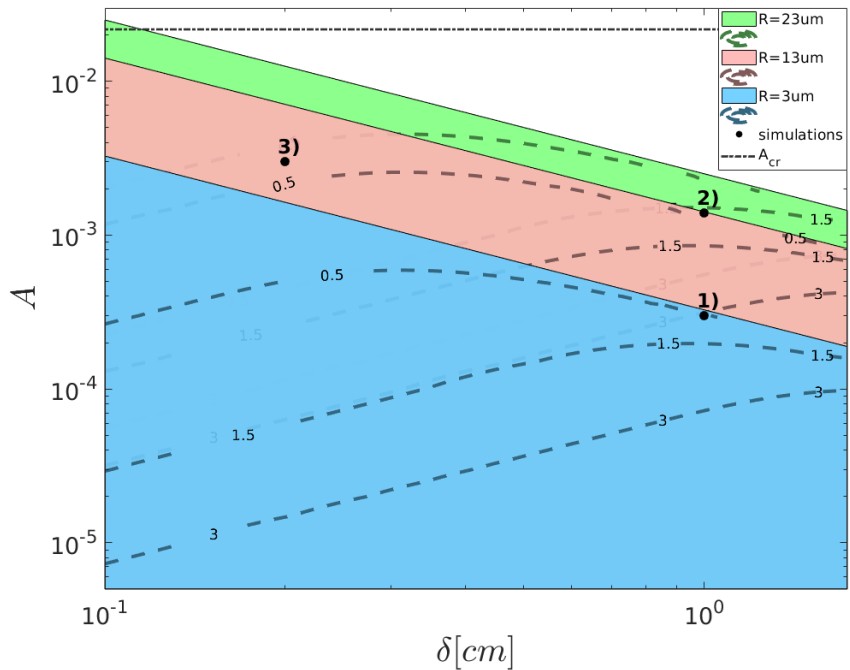

**Figure 5.** Contour plot of stable periodic 2D orbit radius for droplets of radii $R = 3, 13, 23\ \mu m$ covering the experimental range on August 27th. Selected ranges of vortex parameters: vortex core size $\delta$ and strain $A$ are in x an y axes, respectively. Overlapping (blue on a top, then pink and green) colored surfaces match subspaces in which stable periodic 2D orbit exist for droplet radius given by its colour. Dashed lines are contour plots of solutions of chosen void sizes:0.5 cm ,1.5 cm, 3 cm. Black points represent parameters set for simulations as described in Sect.5.

constant in Eq. 1 is in the range $m \in [3.5, 24]$. It means $\delta \in [0.18, 1.20]$ cm for August 27th measurements. Secondly, the droplet trajectory curvature radius is approximated by the periodic orbit radius which is a solution of Eq. 5. For this reason, solutions of Eq.5 are presented in Fig.5 for various representative vortex parameters. Every color represents one of droplet sizes: $R = 3, 13, 23\ \mu m$ chosen to be within the experimental range for August 27th (see Table 1). Overlapping colored surfaces
5   match regions in which solutions can exist (what corresponds to the condition $St < St_{cr}$). Dashed lines are contour plots of solutions of chosen (close to experimental) void sizes: 0.5 cm, 1.5 cm, 3 cm. Using the information presented in this plot, the analysed strain parameter was further limited, from $A < A_{cr}$ down to $A \in [10^{-4}, 8 \cdot 10^{-3}]$.

## 5   Mie scattering influence on particle imaging

Cloud voids were measured by recording light scattered by droplets within an illuminated laser-light sheet. The ability to
10   visualize a void therefore depends not only on the positions and number density of cloud particles, but on the scattering properties of the particles. In this section we consider this optical perspective of the measurement problem.

The Mie scattering theory is a rigorous mathematical theory describing the problem of elastic scattering of light by a dielectric sphere of arbitrary size and homogeneous refractive index in the case in which a sphere size is similar to or larger than the wavelength of the incident light. It shows a complex angular and particle size dependency of the scattered light intensity (van de Hulst, 1957). Thus brightness of images of laser light scattered by polydisperse set of droplets is not expected to be monotonic
with the droplet size. The reasons are described in next paragraphs.

Firstly the camera sensor pixel responds with a signal registration only if it receives an amount of energy exceeding a certain threshold. Light scattered by a particle passes through the optics and undergos some transformations. What is more, the particle image on the sensor is characterized by the internal intensity distribution (diffraction pattern). Image size depends on particle size, optics magnification, position with respect to the focus and other factors (see Olsen and Adrian (2000)).

Secondly the scattered light intensity at an arbitrary angle depends nonlinearly on particle size. A larger particle can give a lower scattered intensity than a smaller one, or there may be several order of magnitude difference in intensity between particles differing by one order of magnitude in size.

For the purpose of our simplified analysis, the following assumptions are made:

- One particle image is recorded by one pixel,

- the signal received by a pixel changes linearly with incident light intensity only,

- each particle is in focus and its image size and depends linearly on the particle size,

- the experiment in clouds was set up to allow best visualisation of maximal number of particles possible.

In order to compare results of the simulations with the measurements a procedure of droplet size and colour scaling is proposed. Calculation of the Mie scattering intensity is performed with the help of an algorithm that was described in Bohren and Huffman
(2007). The scattering angle corresponds to $40 \deg$. In the size range of cloud particles the light intensity has a general growing tendency, but it is still strongly nonlinear. There are 3 orders of magnitude difference between particles of 1 $\mu$m and 30 $\mu$m radius. Relative intensity is calculated on this basis. Next, the brightness scaling is made. It assumes that experiment was set up to enable visualisation of 95% of particle size spectrum. The particle size at which the cumulant of the particle size distribution reaches 95% was calculated. Particles larger than this size have get brighness equal to 1 in the plots. Brightness for the other
particles scales linearly with relative scattered light intensity. To mimic camera sensitivity there is a threshold below which particles get brighness 0. In the plot with white background the relation is opposite, so the brightest particles are black, and the least bright are white. This color scaling was used in the Sect.6 for numerical simulation plots.

## 6   Numerical Simulation Results

The hypothesis of cloud void appearance was verified by numerical simulations. To imitate processes occurring in real vortex
tubes in clouds and examine the effect exerted on a droplet field by the presence of a vortex, a cylinder-shaped domain was chosen. At $t = 0$, the domain was filled uniformly with a given number concentration $n$ of droplets. Droplets leaving the

**Table 5.** Sets of vortex and particle parameters chosen for numerical simulation. $St$, $Sv$, $Fr$ are mean values. Only sets 1) and 2) give the visual effects of the cloud void.

| | $\delta$ [cm] | $\theta$ | $A$ | $A_{max}$ | $\tau_f$ [s] | $St$ | $Sv$ | $Fr$ |
|---|---|---|---|---|---|---|---|---|
| 1) | 1.0 | $3\pi/8$ | 0.0003 | 0.0017 | 0.0020 | 1.23 | 0.0034 | 360 |
| 2) | 1.0 | $\pi/4$ | 0.0014 | 0.0017 | 0.0093 | 0.26 | 0.0159 | 17 |
| 3) | 0.5 | $3\pi/8$ | 0.0030 | 0.0030 | 0.0050 | 0.49 | 0.0223 | 22 |

simulation domain were removed, and no interaction between droplets was imposed. Initial positions of the new droplets at $t > 0$ were randomized on the cylinder surface to obtain conditions "outside the cylinder" of homogeneous spatial distribution with the same $n$. The initial velocity of these droplets was adjusted to the radial inflow velocity. The simulation domain size was chosen to be capable of showing phenomena at scales larger than standard experimental cloud void size, such that $Z =$

20 cm long and $D = 5$ cm in radius. Sensitivity to different values of $D$ was examined, and no significant dependence on void creation was noticed. Droplet trajectories were calculated by solving numerically Eq.3 in the Burgers vortex velocity field. Droplet number concentration within the domain is almost constant and the pattern does not change. The droplet radius distribution matches the experimental distribution from the 27th of August (see Table 1 and Fig.2 for comparison). Since droplets do not interact with each other and results do not depend on droplet concentration, $n = 10$ cm$^{-3}$ was chosen as a

compromise between computational costs and visibility purposes. After a few seconds each simulation becomes steady. A $40°$ scattering angle was chosen for calculation of Mie effect in the post-processing of simulation results.

The parameter sets for three representative simulations are presented in Table 5. Two of them, 1) and 2) may result in the presence of a round or close to round void around the vortex axis. Simulation 3) does not show anything close to void creation or any other persistent pattern formation.

Figure 6 presents a 3D views (panels a and b) with a 2D cross-sections (c and d) at the last, quasi stationary stage of simulations 1) and 2). In the database (Karpinska et al., 2018) one may find 2D and 3D animations from the simulations. Animations "ms03" and "ms04" correspond to set 1), "ms05" and "ms06" to set 2), "ms07" and "ms08" to set 3).

The droplet spatial distribution in and around presented voids show signs of clustering and segregation. Standard quantitative indicators of these phenomena (radial distribution function, fractal dimension, Voronoi diagram, segregation lenght) designed

for homogeneous isotropic turbulence in our case would be difficult to interpret so another approach is proposed.

The droplet size distribution was divided into five bins in such a way that the bins are equal in droplet number. Number concentration $n$ with respect to the distance from the vortex axis $r$ was plotted in panels e) and f) of Fig. 6. The ploted $n$ values represent the mean over 50 succesive time instants. Different shading colors represent contributions from different size groups, so the darkest plot represents smallest droplets - first size group, the brighter represents the first and the second size group

together etc. If the distribution of droplets in the domain was homogeneous it would be plotted as horizontal, parallel, equally spaced lines. Red line plots in the same panel present droplet mean radius $\langle R \rangle$ and droplet mean visible radius $\langle S \rangle$ versus distance from the vortex axis. The visible radius reflects droplet relative sizes in the camera image estimated by including Mie scattering and the camera threshold sensitivity influence as defined in Sec.5. In the absence of sorting both plots would

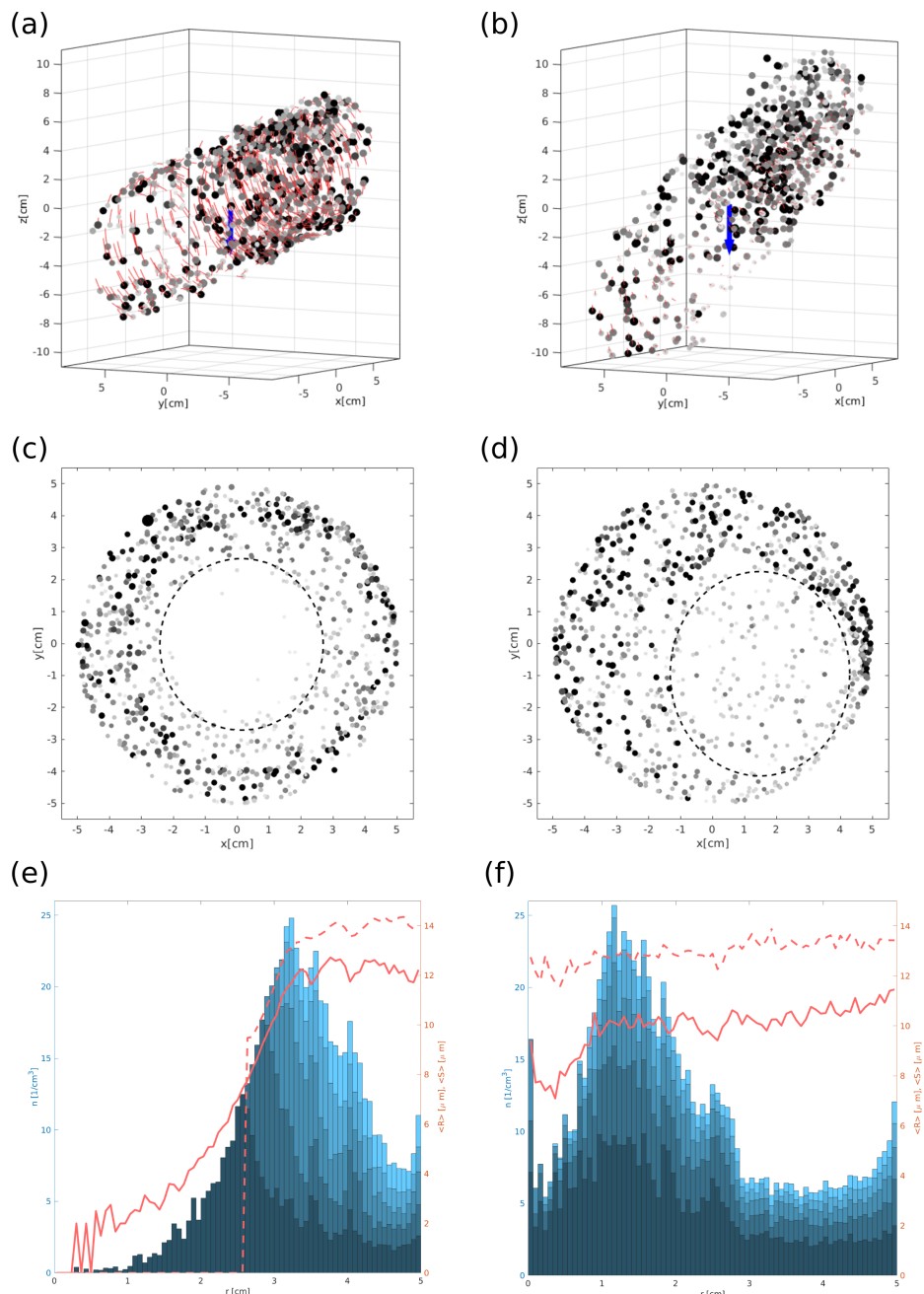

**Figure 6.** Positions of droplets in simulations 1) (left column) and 2) (right column) as described in Table 5 in perspective view (panels a and b) and in $4\ cm$-thick central slice projected on a plane perpendicular to the vortex axis (panels c and d). The blue arrow shows the direction of gravity. In panels a-d, droplet size and colour are scaled respectively to the rules described in Sec.5. The red tracks behind the particles in panels a and b reflect the last $\Delta t = 0.0055$ s of the droplet trajectory (for clarity every 10th droplet is drawn). The dashed circles in panels c and d reflect the approximate shape and size of the visible void in the droplet field. Panels e and f present: the number concentration of droplets $n$ with different blue shading colors representing the contribution from different size groups - left axis, droplet mean real radius (solid red line) and mean visible radius (dashed red line) - right axis, all with respect to the distance from the vortex axis $r$.

approach one horizontal line.

Panels e and f of Fig. 6 show that droplets around the void are nonhomogeneously distributed and segregated by size in space. The most inner part of the figure in panel e) is almost devoid of droplets. The larger distance from the axis the bigger
the particles circling around the void. Animations of motion clearly show that this is caused by limit cycle attraction, which radius apparently grows with droplet size. This is pronounced in panel c as well. The visible void radius is around 2.5 cm, whereas "real" (no droplets) void radius is estimated to be 1 cm. Panel f of Fig. 6 shows the more complex situation: the inner part of the vortex is not empty, but the droplets are smaller (less visible) then those further away from the axis. For increasing visibility threshold, the visible void size increase in both cases.

Values of $St$, $Sv$ (mean of polydisperse particle set) in simulated vortices differ essentialy from values in Table 1. Estimations of $St$ and $Sv$ in 3D real turbulence flow are based on a global value of the Kolmogorov timescale. The calculation in the simulation uses vortex model characteristic times. Local parameter values of single vortex in intermittent turbulent flow can be completely different than global flow characteristics.

In summary, there are two possible factors together creating the visible effect of the void. The first one is collective droplet
dynamics: the majority of the droplets move on helical trajectories, being attracted in 2D space by the limit cycle or by a stable point near the axis. Large droplets may be slowly attracted by their equilibrium point far from the vortex axis in 2D space, but in the course of attraction they circle around the axis. At the same time, a significant ratio of the characteristic timescale of motion in the plane perpendicular to vortex axis with respect to motion along the vortex axis is needed. Secondly, segregation of particles with respect to the distance from the vortex axis can influence visible void size due to Mie scattering effects. Even
if the circulation is not strong enough to displace the smallest particles far from the vortex axis it is possible that their images are not recorded by the camera. Therefore visible void size can depend vitally on imaging capability.

## 7   Discussion and Conclusions

Visualizations of cloud droplets by means of laser sheet photography performed at the Schneefernerhaus observatory revealed
the presence of voids - holes in the form of curved elongated cylinders with a radius of a few centimeters. The possibility of such cloud voids or "Swiss cheese" cuts in clouds was suggested by former studies of the Stokes motion of particles in idealized vortices. The original work of Marcu et al. (1995) elaborating on single particle motion in a Burgers vortex was adopted to explain the behavior for a specific scenario: polydisperse cloud droplets inside a vortex, with cloud particle and vortex properties scaled to the ranges consistent with our cloud microphysical and turbulence observations.

Using information on cloud droplet size distributions and turbulence parameters collected in the course of observations, as well as literature discussions on vortex tubes in turbulent flows, we have shown that the cloud voids observed under the experimental conditions were likely a result of the presence of relatively thin yet long vortex tubes. Approximate theoretical conditions of void creation were proposed and vortex circulation was shown to be the parameter of the greatest importance in the conditions

formulation. The calculations are consistent with the observation that voids are present under some conditions and not under others. Comparison of the modeled and experimental voids led to the conclusion that properties of the Mie scattering are crucial for reproducing the proper size and shape of cloud voids observed by laser imaging.

Our analysis shows, that existence of Burgers-like vortices in the measurement conditions are likely and they may explain the observed phenomenon. This finding, if confirmed in clouds far from the atmospheric surface layer, might help to better understand the effect of high Reynolds number turbulence on clustering, size segregation and probably collisions of cloud droplets. In the literature, several perspectives on the clustering mechanism were presented, to name just a few: Maxey (1987), Coleman and Vassilicos (2009), Falkovich and Pumir (2007),Gustavsson and Mehlig (2011). The paper of Gustavsson and Mehlig (2016) presents a thorough review of research on clustering. It is recognized that consideration of droplet motion within a single vortex as a representative of coherent dissipative structures expected to exist in turbulent flows, is a strong simplification in comparison to the cited works. The use of simplified models can be accepted at a semi-quantitative level to understand what factors influence void formation, but cannot replace statistical analysis of the droplet spatial distribution in fully three-dimensional turbulent flow. There are very few numerical simulations of particle-laden 3D turbulence that capture the range of scales relevant to this problem, however, i.e., from the large scales that feed the intermittent formation of intense vortices, to the dissipation scales relevant to actual droplet clustering. Thus, it is reasonable to explore the essential physics of the problem using vortex models; for example, the research conducted for non-sedimenting particles in a simple vortex by Ravichandran and Govindarajan (2015) and Deepu et al. (2017) suggests that fixed point attraction and caustics formed by limit cycle attraction strongly increase the clustering and collision probability of particles near single and multiple vortices. This fact should become very distinct motivation for investing in both experimental and theoretical research aiming at thorough quantitative characterization of cloud void events.

*Code and data availability.* Numerical simulation code available on demand. Data repository containing experimental movies and animations of simulations is retrieved from: https://www.researchgate.net/publication/328429794_Data_supporting_the_paper_Turbulence_induced_cloud_voids_observation_and_interpretation.

*Author contributions.* Raymond Shaw, Holger Siebert and Eberhard Bodenschatz designed the experiment. Szymon Malinowski formulated the aim for theoretical and numerical analysis. Steffen Risius, Tina Shmeissner, Raymond Shaw, Holger Siebert, Hengdong Xi, Haitao Xu Jonathan Bodenschatz and Eberhard Bodenschatz provided resources and carried out the experiment. Tina Shmeissner, Steffen Risius, Haitao Xu, Eberhard Bodenshatz maintained and synthesized experimental data. Szymon Malinowski and Eberhard Bodenschatz acquired financial support. Katarzyna Karpińska and Szymon Malinowski developed the numerical model. Katarzyna Karpińska analysed the experimental data, conducted theoretical analysis, designed and implemented the code for numerical analysis, conducted validation and verification of numerical results, wrote the original draft of the paper. Katarzyna Karpińska and Steffen Riusius provided visualisation of the results. Szymon Malinowski was responsible for supervision. Katarzyna Karpinska, Szymon Malinowski, Jakub Nowak, Steffen Riusius, Raymond Shaw and Holger Siebert edited the original draft to prepare it for submission.

*Competing interests.* No competing interest are present.

*Acknowledgements.* We are grateful to Mr. Markus Neumann and the staff at UFS for their technical help at UFS and the Bavarian Umwelt-ministerium for the financial support of the station. Financial support from Max Planck Society, Deutsche Forschungsgemeinschaft (DFG) through the SPP 1276 Metström, the EU COST Action MP0806 Particles in Turbulence and through the US National Science Foundation

5   (NSF grants AGS-1026123 and AGS-1754244) are gratefully acknowledged. Theoretical analysis and simulations were possible due to the support from the Polish National Science Centre (grant 2013/08/A/ST10/00291). We thank Marta Wacławczyk and Liang Ping Wang for the comments on the manuscript.

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
