# Peer review of "Turbulence Induced Cloud Voids: Observation and Interpretation"

_Atmospheric Chemistry and Physics, 2018_

## Referee Comment (RC1) · Anonymous Referee #1 · 16 Nov 2018

**Report on "Turbulence Induced Cloud Voids: Observation and Interpretation" (acp-2018-1049) by K. Karpinska et al.**

This paper describes a theoretical explanation of physical mechanism of generating voids in clouds and some results of kinematic simulation. First, the authors explain the measurements of droplets inside cloud made at the top of German Alps. Then the size distribution of droplets is presented and the voids in the cloud droplet distribution are reported in figures and videos, which are very interesting. Next, the authors study the physical mechanism of void formation. Basic idea is to use the idea that an inertial particle tends to be expelled from the core of Burgers vortex. Theoretical analysis for this physics had already been done by Marqu et al. cited in the reference, and the present paper applies, in its essence, simply the results to the void formation. In order to see whether such interpretation is the case, the authors perform the kinematic simulation, to numerically integrate a set of equations of particles with different sizes in the flow field generated by the Burgers vortex. For two cases of parameters among three cases, the distribution of droplets is found to generate voids, which is consistent with Fig.5 of the theoretical prediction. Finally they argue the physical mechanism from the view points of the present analysis and visualization of voids.

The parts of the experiment and kinematic simulation are, I think, new and it is an idea to use the analysis in terms of Burgers vortex flow to explain the void formation. However, the part regarding the equilibrium positions and their stability is poorly written and hard to follow. The pages of p. 9-11 use the results of Marqu et al. without definition of some quantities, such as $\phi$ and $r_0$. While the authors present the parameter range for stable periodic orbit in Fig.5, the kinematic simulations were done only for three points in the figure. Examination of the relevance of the simulation results by comparing to the theoretical results is not enough to convince the physical reasoning of void formation. Also in the discussion, the authors argue Mie scattering as one possibility of observing the void, but this is a speculation and lacks fundamental data or analysis. In conclusion, the present manuscript needs considerable revision.

Technical points are listed below.

1. p.8, 4th line. the authors write $A$ as the strain parameter. I think, this is not appropriate. $\Gamma$ is the circulation of the Burgers vortex so that $A = 1/Re$ where $Re$ is the Reynolds number of the vortex.

2. p.9. What are definitions of $r_0^*$ and $\phi(r_0^*)$ ? The paragraph of eqs. (7) and (8) without explanation does not make arguments useful.

3. p.11 How is $A_{cr}$ determined?

---

## Referee Comment (RC2) · Anonymous Referee #2 · 20 Nov 2018

Review report on "Turbulence Induced Cloud Voids: Observation and Interpretation" by K. Karpinska et al.

Authors report about an experimental investigation of the behaviour of water droplets measured in cloudy air at a mountain-top station. In particular, they focus on cloud voids, that is spatial regions which are devoid of droplets. To explain the observed phenomenon, they perform a numerical study of a model of inertial particles moving in a Burgers vortex (Marcu et al 1995), under the action of drag and gravity forces. On the basis of these results, with model parameters matched to those of the experiments, authors draw the conclusion that cloud voids observed under the experimental conditions were very likely a result of the presence of relatively thin yet long vortex tubes.

The most interesting part of this work is the observation of the phenomenon: cloud

voids at the centimetre scale tell us that cloud droplets do not distribute homogeneously in space and that inhomogeneities take place also at scales much larger than the estimated Kolmogorov scale of the flow.

As for the analysis and interpretation of results, I find it weak for the following reasons. To me, the application of a simplified model like the Burgers vortex one is meaningful if we can learn something new about the dynamics of inertial particles in turbulent flows. However this does not seem the case. Indeed a detailed theoretical analysis for the motion of inertial particles in these vortical structures had already been done by Marcu et al.; the present paper reproduces one of possible scenarios of the model with suitable parameters, without adding new knowledge.

Moreover, since the work of Marcu et al., there has been a considerable amount of research about statistical characterisation of inertial particle spatial distribution in turbulent flows. Many different analysis in terms of deviations from Poisson distribution, Radial Distribution Function or Voronoi diagrams, have been applied to the case of polydisperse solutions also (see e.g. 2012 New J. Phys. vol. 14, 095013). Let me stress that voids appear at scales which are about 20-40 Kolmogorov scales, so distances at the edge between the dissipative and the inertial ranges of turbulence, where the strongest intermittent fluctuations take place.

To summarise: turbulence at small scales is very different from a superposition of Burgers vortices, the use of simplified models can be accepted at a qualitative level to make things clearer but can not replace statistical analysis. Since the observations are interesting and worth of publication, I suggest the authors to perform additional work and to make a considerable revision of the manuscript.

Detailed comments 1) Starting from the Abstract and then few times in the paper, authors speak about sorting effect. What is it? Can they state clearly what is the sorting effect and how it is different/similar to the preferential concentration effect?

2) In Section 2, the authors should detail what are the Mie scattering visual effects.

These are often called as possible co-responsible for the creation of voids, but there are not data or analysis related to them. Have these effects been observed in laboratory experiment? Can we reproduce similar conditions? Either the authors better detail what they have in mind, or it is better to simply mention the problem once.

Also in Section 2, they mention that the "velocity structure functions were calculated using Taylor's frozen-flow hypothesis, and the energy dissipation rates were determined using inertial range scaling". Can they show these data to see the extent of the inertial range both for the temporal and the spatial scales?

3) At the end of page 4, the authors mention that they selected 27 voids for further analysis. Are these statistically equivalent? Can the authors perform a statistical analysis of the way droplets distribute in space? See comments above.

4) Table I should be enriched with turbulent flow parameters such as the value of the Taylor scale Reynolds number $Re_{\lambda}$, the value of the Kolmogorov scale $\tau$, estimate for large-scale eddy-turn-over-time $T_L$ and correlation length $L$, the expression used for St and Sv.

5) Pages 8 and beyond: I would suggest that values such as $A_{cr} = 0.02176$ or $r_i=2.1866$ to be put in a table, they have no special physical meaning (only within the Burgers vortex model).

6) Section 4 is not clear and moreover as the authors specify: "Different scenarios of particle motion determined by above stability conditions were shown in Fig.4 in Marcu et al. (1995). Fig. 4 here presents a simplified illustration of one of the scenarios: three equilibrium points, unstable point near the axis, stable point far from the axis, and droplets rotating around the vortex center." How are the other scenarios excluded? What is special in the one chosen?

Here the whole analysis can also be made much shorter, and summarised in terms of the very natural rough conclusions mentioned at the end of page 11. Details can be

moved in an appendix.

7) Authors want to reproduce observations of August 27: from figure 1 it is the day with a very broad radii distribution. Have they used this shape to initialise the numerical simulations? Or can they superpose the shape they used to the experimental one? This is not clear from the text and the sentence "A semi-Gaussian distribution of droplet radii cut off at R = 1.5$\mu$m was chosen for simulations to match the experimental values from the 27th of August (see Table 1)" without further details does not clarify the point.

8)The work reports about void/clustering effects of inertial particles in turbulent flows but many relevant papers for the topic are not cited. To mention just a few: Collins and Keswani, 2004 New Journal of Physics 6 (1), 119; Bec et al. 2007 Phys. Rev. Lett. 98, 084502; Monchaux, Bourgoin, and Cartellier, 2010 Phys. Fluids 22, 103304.

9) A number of typos are present.

---

## Referee Comment (RC3) · Anonymous Referee #3 · 26 Nov 2018

**Comments on 'Turbulence induced cloud voids: observation and interpretation'**

This is an interesting article with some novel field data and numerical results which have been interpreted with reference to existing theoretical analysis. The extent to which this analysis is strictly applicable to real clouds is perhaps moot but it nevertheless provides a useful starting point and appropriate orders of magnitude. I would recommend publication but found aspects of the paper, particularly the summary of the analysis (§3) and its application (§4) confusing. I have a number of detailed points, mostly minor, which I would like the authors to consider especially with a view to improving the clarity of their arguments. I have also made a number of suggestions for improving the written English.

**Detailed comments**

p.3, l. 29 It's not clear what 'They' in the sentence 'They were smaller ...' is referring to here: 'Swiss cheese' or 'cloud holes'?

p.6, eq. (2) Could the authors provide a reference for this? Or at least some more motivation?

p.9, l. 2 What is $r_0$? Does it have any physical relevance? It would be helpful if $\phi(x)$ were defined so that its dependence on the parameters in the problem is made clear; it has an important role in the analysis.

p.9, l.12 I didn't understand the sentence beginning 'Particle motion ...' especially 'separated from the motion 2D space'.

p.11, eq. (9) For the linearization of eq. (6) I obtained

$$r^* \approx \frac{Sv}{A}(1 + (4\pi A)^2)^{-1/2}$$

While this differs from eq. (9) for non-zero $A$, it does give me $r^* \approx 4\pi Sv$ in the limit $A \to 0$ in agreement with eq. (9).

p.11, l.16 I didn't understand the first sentence '... splits in parts'. Two parts?

p. 11, l. 23+ Why are the conclusions 'rough'? I didn't follow all of the logic here: as the influence of gravity increases i.e. $g\sin\theta \uparrow$ or the vortex size, $\delta$, increases (or both) then $B$ decreases. Decreasing $B$ means decreasing $A_{\max}$ which is consistent with increasing $\Gamma$ (circulation). My understanding is that $A < A_{\max}$ for void creation. So as either $g\sin\theta$ or $\delta$ (or both) increase it becomes harder for voids to form unless the circulation increases appropriately. So I agree with the first conclusion so long as increasing minimum circulation equates to decreasing $A_{\max}$.

A further consequence of decreasing $B$ is decreasing $St_1$ and $\Delta St$ since the latter is proportional to $B^{1/3}$. This implies that, for fixed $\tau_f$ (circulation), the minimum particle size decreases with decreasing $B$ as the second point suggests though it is not written very clearly. Since $\Delta St$ decreases with decreasing $B$ this suggests the range of particle size decreases with decreasing $B$ as suggested by the latter part of the third point. But decreasing $B$ corresponds to increasing $\Gamma$ which seems to contradict the first part of the third conclusion. Perhaps I have missed something in the analysis of §§3 & 4; I would welcome more explanation.

p.12, l.5 The last part of this sentence doesn't read well.

p.12, §6 Which equations are the numerical simulations solving? A Burgers vortex or the Navier-Stokes equations?

p. 12, l. 32 $Z$ and $D$ should be defined: are they imposed on the simulation or simply typical values?

p.16, l.14 My understanding of the analysis of §§3 & 4 is that $A < A_{\max}$ for void creation yet here you are saying the opposite.

---

## Referee Comment (RC4) · Anonymous Referee #4 · 27 Nov 2018

The paper presents field experimental observation of the so defined "cloud voids" and attempts to provide physical mechanism for this phenomena based on the simple model of inertial particle expulsion from strong, thin coherent vortices, using burger Burger vortex as model.

The field observation of the cloud voids is in itself new and interesting and worth disclosing. However, the link to expulsion from coherent vortices is, although plausible, not as convincing as suggested in the Abstracts and Conclusion of the paper. I would recommend a somewhat weaker conclusion unless more work is done to strengthen this interpretation. In relation to this and other weaknesses, below are my detailed comments:

1. According previous works such as the cited Mouri et al (2000), coherent intense

[Figure]

vortices are observed to have diameter not more than about 20 times the Kolmogorov microscale (as reported by Mouri et al.). This is consistent with the authors' own survey as they wrote "proportionality constant in Eq. 1 is in the range m âĹĹ [3.5, 24]". Their observed void size (centimeters) in this report is however about a hundred times the corresponding Kolmogorov length (see Table 1.). This calls for an explanation. E.g. are larger vortices expected in atmospheric conditions?

2. As the authors stated, these coherent vortices are "severely intermittent". What could be said on the prevalence of the observed voids at the site? And could this be reconciled with known intermittency, perhaps in terms of "large-scale organization of the small-scale intermittent structures" ?

3. Ideally, it would be very helpful, if the velocity or vorticity field around the observed void is also concurrently estimated if not measured. If that is not the case, it would be helpful to establish the presence of intense vortices, long lived enough and having similar diameters under similar conditions, at the site.

4. The Stokes number used in simulation 1) is St=1.45. I was hoping to see St similar to the experimental value (similarly for S_v) for a better comparison (is this not the purpose, why?, in any case I recommend that). In relation to this, I don't know if it is meaningful to claim "visible void radii are rather âĹij 2-2.5 cm, which seems close to the experimental values" (line-4, pg.16) when parameters are not matched.

5. I have difficulties understanding Figure 5 and the corresponding explanation in the text. In particular, I am not sure how to interpret the dashed-lines. Better exposition is welcomed here.

6. Line-15, pg 16: "Comparison of the modeled and observed voids led . . ..". observe refers to the field data or the "visible void" in the simulation?

7. Line-4, pg.16 : "0-1.5 cm; however, . . ..". Is zero a typo here?

[Figure]

2018.

---

## Referee Comment (RC5) · Anonymous Referee #3 · 27 Nov 2018

[referee-annotated manuscript omitted]

---

## Author Comment (AC1) · 6 Feb 2019

**Authors response to reviews**
**Turbulence Induced Cloud Voids: Observation and Interpretation**

Katarzyna Karpinska[1], Jonathan F.E. Bodenschatz[2], Szymon P. Malinowski[1], Jakub L. Nowak[1], Steffen Risius[2], Tina Schmeissner[4], Raymond A. Shaw[3], Holger Siebert[4], Hengdong Xi[2], Haitao Xu[2], and Eberhard Bodenschatz[2]

[1]Institute of Geophysics, Faculty of Physics, University of Warsaw, Warsaw, Poland
[2]Max Planck Institute for Dynamics and Self-Organization, Goetingen, Germany
[3]Michigan Technological University, Houghton, Michigan, USA
[4]Leibniz Institute for Tropospheric Research, Leipzig, Germany

**Review 1**

This paper describes a theoretical explanation of physical mechanism of generating voids in clouds and some results of kinematic simulation. First, the authors explain the measurements of droplets inside cloud made at the top of German Alps. Then the size distribution of droplets is presented and the voids in the cloud droplet distribution are reported in figures and videos, which are very interesting. Next, the authors study the physical mechanism of void formation. Basic idea is to use the idea that an inertial particle tends to be expelled from the core of Burgers vortex. Theoretical analysis for this physics had already been done by Marqu et al.cited in the reference, and the present paper applies, in its essence, simply the results to the void formation. In order to see whether such interpretation is the case, the authors perform the kinematic simulation, to numerically integrate a set of equations of particles with different sizes in the flow field generated by the Burgers vortex. For two cases of parameters among three cases, the distribution of droplets is found to generate voids, which is consistent with Fig.5 of the theoretical prediction. Finally they argue the physical mechanism from the view points of the present analysis and visualization of voids.

The parts of the experiment and kinematic simulation are, I think, new and it is an idea to use the analysis in terms of Burgers vortex flow to explain the void formation. However, the part regarding the equilibrium positions and their stability is poorly written and hard to follow.

The pages of p. 9-11 use the results of Marqu et al. without definition of some quantities, such as $\varphi$ and $r_0$ . While the authors present the parameter range for stable periodic orbit in Fig.5, the kinematic simulations were done only for three points in the figure. Examination of the relevance of the simulation results by comparing to the theoretical results is not enough to convince the physical reasoning of void formation. Also in the discussion, the authors argue Mie scattering as one possibility of observing the void, but this is a speculation and lacks fundamental data or analysis. In conclusion, the present manuscript needs considerable revision.

**We substantially edited the manuscript, in particular the part regarding the equilibrium positions and their stability**

**with respect to void creation in order to improve presentation quality. We also substantially modified the discussion concerning Mie scattering in order to discuss its effect on the collected data or images in a more clear way.**

Technical points are listed below.

5  1. p.8, 4th line. the authors write A as the strain parameter. I think, this is not appropriate. $\Gamma$ is the circulation of the Burgers vortex so that $A = 1/Re$ where Re is the Reynolds number of the vortex.
   **$A$ is called "nondimensional strain parameter" in agreement with the name used in the work Marcu et al. (1995). It is true that it is an inverse of vortex Reynolds number and this information was added in the text.**

   2. p.9. What are definitions of $r_0$ and $\varphi(r_0)$ ? The paragraph of eqs. (7) and (8) without explanation does not make argu-
10     ments useful.

   **$r_0^*$ is just the notation for arbitrary solution of the Eq.6. The function $\varphi(r_0^*)$ has no parameters. On the basis of this suggestion the text was modified to expressed the intention expicitely.**

   3. p.11 How is Acr determined?
15   **$A_{cr}$ is defined at p.9. Its value is obtained numerically. For $A = A_{cr}$ the equillibrium curve has a horizontal slope at the inflection point.**

**Review 2**

Authors report about an experimental investigation of the behaviour of water droplets measured in cloudy air at a mountain-top station. In particular, they focus on cloud voids, that is spatial regions which are devoid of droplets. To explain the observed
20  phenomenon, they perform a numerical study of a model of inertial particles moving in a Burgers vortex (Marcu et al 1995), under the action of drag and gravity forces. On the basis of these results, with model parameters matched to those of the experiments, authors draw the conclusion that cloud voids observed under the experimental conditions were very likely a result of the presence of relatively thin yet long vortex tubes. The most interesting part of this work is the observation of the phenomenon: cloud voids at the centimetre scale tell us that cloud droplets do not distribute homogeneously in space and that
25  inhomogeneities take place also at scales much larger than the estimated Kolmogorov scale of the flow.
As for the analysis and interpretation of results, I find it weak for the following reasons. To me, the application of a simplified model like the Burgers vortex one is meaningful if we can learn something new about the dynamics of inertial particles in turbulent flows. However this does not seem the case. Indeed a detailed theoretical analysis for the motion of inertial particles in these vortical structures had already been done by Marcu et al.; the present paper reproduces one of possible scenarios of
30  the model with suitable parameters, without adding new knowledge.
Moreover, since the work of Marcu et al., there has been a considerable amount of research about statistical characterisation of inertial particle spatial distribution in turbulent flows. Many different analysis in terms of deviations from Poisson distribution,

Radial Distribution Function or Voronoi diagrams, have been applied to the case of polydisperse solutions also (see e.g. 2012 New J. Phys. vol. 14, 095013). Let me stress that voids appear at scales which are about 20-40 Kolmogorov scales, so distances at the edge between the dissipative and the inertial ranges of turbulence, where the strongest intermittent fluctuations take place. To summarise: turbulence at small scales is very different from a superposition of Burgers vortices, the use of simplified models

5    can be accepted at a qualitative level to make things clearer but can not replace statistical analysis. Since the observations are interesting and worth of publication, I suggest the authors to perform additional work and to make a considerable revision of the manuscript.

**We agree with te reviewer that we use a simplified Burgers vortex model exploited by Marcu to explain the exper-**

10    **imental data. However, the original work of Marcu is just one of many theoretical models. First of all we used it to explain behavior of polydysperse particles inside a vortex and this had not been done before. What is even more important, we validate theoretical approach by an observation of a particular phenomenon - cloud voids.**
**We agree that turbulence at small scales is very different from a superposition of Burgers vortices. The use of simplified models can be accepted at a qualitative level to make things clearer but can not replace statistical analysis of droplet**

15    **spatial distribution in 3D turbulent flow. Nonetheless there are just a few analysis specific of cloud droplets in small scales, no 2D or 3D data on cloud droplets spatial distributions in the domains of void size and larger, not to mention simultaneous measurement of droplet position and size. Our analysis shows, that existence of Burgers-like vortices in the measurement conditions are likely and they may explain observed phenomenon, which adds certainly new knowlegde. We make an effort to improve the manuscript and present our intensions in more clear way.**

Detailed comments

1. Starting from the Abstract and then few times in the paper, authors speak about sorting effect. What is it? Can they state clearly what is the sorting effect and how it is different/similar to the preferential concentration effect? **We define**

25      **clustering of particles as inhomogeneous distribution of particles in space. Preferential concentration (or preferential sampling) is specific kind of clustering: particles sample certain regions of the flow, so there is correlation between particle spatial distribution and spatial distribution of a flow characteristic/s. The term "sorting" was in fact used interchangeably to "segregation" which describes polydisperse statistics: spatial distribution of different sized particles are anticorrelated. To make our point about clustering and sorting in vortex tube clear these**

30      **explanation was incorporated in the manuscript.**

2. In Section 2, the authors should detail what are the Mie scattering visual effects. These are often called as possible co-responsible for the creation of voids, but there are not data or analysis related to them. Have these effects been observed in laboratory experiment? Can we reproduce similar conditions? Either the authors better detail what they have in mind, or it is better to simply mention the problem once.

[Figure]

**Figure 1.** Relative scattered light intensity versus angle for a water droplet in the air. Colors correspond to different droplet radii.

To clarify this point we added a paragraph about Mie scattering role in particle imaging and its potential influence on void observation. Mie scattering theory (van de Hulst, 1957) was validated experimentally with the accuracy far beyond our use (Harris, 1969). There is as well a great extent of literature devoted to Mie theory and its applications in experiments using scattering on particles both in the laboratory flows and in the atmosphere (see for example Dyer et al. (2006), Graßmann and Peters (2004), Fischer (2017)). The exact impact on particle tracking and sizing in imaging experiments performed in atmospheric conditions is an area of current research. We are definitely not able to reproduce in the laboratory conditions similar to those described in the paper in the laboratory. Hence we cannot measure directly the Mie scattering influence on void creation. What we know is that the intensity of scattered light is nonlinear with particle size and the angle of scattering, what we present in Fig.1 here. Nonetheless we formulate the hypothesis of its great influence on vortex finall image with the aim to sensitize to potential bias it may cause in the interpretation of observations.

[Figure]

**Figure 2.** Turbulence masurement second order velocity structure functions during the cloud void event of 27th August, from MPIDS sonic sensors (sampled at 10Hz)

Also in Section 2, they mention that the "velocity structure functions were calculated using Taylor's frozen-flow hypothesis, and the energy dissipation rates were determined using inertial range scaling". Can they show these data to see the extent of the inertial range both for the temporal and the spatial scales?

**We provide the structure functions plots made for data aquired during cloud void events in Fig.2 and 3 here.**

5        **Generally the experimental conditions were similar to these discussed in detail in Risius et al. (2015), and presented in Fig. 10 there. Obviously, lower part of the inertial range is not resolved there. Hot wire measurements, resolving smaller scales are discussed in Siebert et al. (2015).**

3. At the end of page 4, the authors mention that they selected 27 voids for further analysis. Are these statistically equivalent?

10        lent? Can the authors perform a statistical analysis of the way droplets distribute in space? See comments above.

**The reason for publishing the observations of voids described in this paper is that they were unique. They were conducted in the brain-storming process and the consequence of this fact is that they lack certain dilligence. The data collected does not certainly allow for analysis of droplets distribution in space. Due to the specific difficulty connected to the measurements conducted in the atmosphere there is no reliable literature data about 2D cloud**

15        **droplets distributions ranging to inertial scales whatsoever. The data collected suggest that turbulence conditions and droplet size distributions were quite steady in the course of the void observations and we are convinced this**

[Figure]

**Figure 3.** Turbulence masurement second order velocity structure functions during the cloud void event of 29th August, from MPIDS sonic sensors (sampled at 10Hz)

**allowes us to compare these voids as they were statistically equivalent. Even if it is not the case the sheer information of their existence, approximate size and a proof they may not be the result of edge effect is a complete novelty and should be published to inspire future more thorough investigations.**

4. Table I should be enriched with turbulent flow parameters such as the value of the Taylor scale Reynolds number $Re_\lambda$, the value of the Kolmogorov scale $\tau$, estimate for large-scale eddy-turn-over-time $T_L$ and correlation length $L$, the expression used for St and Sv. **The Kolmogorov scale $\tau_\eta = \sqrt{\nu\epsilon^{-1}}$ was added to the Table 1. Stokes number and sedimentation parameters were estimated using the expressions $St = \tau_p\tau_\eta^{-1}$, $Sv = \tau_\eta\tau_g^{-1}$, where $\tau_p$ is particle inertial response time, $\tau_g = \eta(g\tau_p)^{-1}$ is a time of sedimentation through the vortex. The adequate explanations are added in the text. We do not have reliable estimates of the other parameters.**

5. Pages 8 and beyond: I would suggest that values such as $A_{cr} = 0.02176$ or $r_i = 2.1866$ to be put in a table, they have no special physical meaning (only within the Burgers vortex model).
   **Values were put in the table as suggested.**

6. Section 4 is not clear and moreover as the authors specify: "Different scenarios of particle motion determined by above stability conditions were shown in Fig.4 in Marcu et al. (1995). Fig. 4 here presents a simplified illustration of one of the scenarios: three equilibrium points, unstable point near the axis, stable point far from the axis, and droplets

rotating around the vortex center." How are the other scenarios excluded? What is special in the one chosen? **Fig.4 was included in the paper for illustrative purposes. We show particle trajectories around equillibrium points of different stability properties in the simplest way without copying the original image. Presenting the chosen scenario fulfills this goal and helps in the discussion.**

7. Here the whole analysis can also be made much shorter, and summarised in terms of the very natural rough conclusions mentioned at the end of page 11. Details can be moved in an appendix.
   **The other reviewers asked for more details in the analysis and that is the reason we decided to leave it in the main text body.**

8. Authors want to reproduce observations of August 27: from figure 1 it is the day with a very broad radii distribution. Have they used this shape to initialise the numerical simulations? Or can they superpose the shape they used to the experimental one? This is not clear from the text and the sentence "A semi-Gaussian distribution of droplet radii cut off at R = 1.5um was chosen for simulations to match the experimental values from the 27th of August (see Table 1)" without further details does not clarify the point.
   **We used the gaussian distribution as presented in Fig 4 here. Havng received this remarks we decided to change in the simulations the gaussian distribution to the one measured.**

9. The work reports about void/clustering effects of inertial particles in turbulent flows but many relevant papers for the topic are not cited. To mention just a few: Collins and Keswani, 2004 New Journal of Physics 6 (1), 119; Bec et al. 2007 Phys. Rev. Lett. 98, 084502; Monchaux, Bourgoin, and Cartellier, 2010 Phys. Fluids 22, 103304.
   **The thorough review of the research on clustering was not our intention. We mentioned only the papers that described different important clustering mechanism. We rephrased it in the conclusions and we hope it is now clear.**

10. A number of typos are present.

**Review 3**

This is an interesting article with some novel field data and numerical results which have been interpreted with reference to existing theoretical analysis. The extent to which this analysis is strictly applicable to real clouds is perhaps moot but it nevertheless provides a useful starting point and appropriate orders of magnitude. I would recommend publication but found aspects of the paper, particularly the summary of the analysis (§3) and its application (§4) confusing.

I have a number of detailed points, mostly minor, which I would like the authors to consider especially with a view to improving the clarity of their arguments. I have also made a number of suggestions for improving the written English.

[Figure]

**Figure 4.** Droplet size distributions measured with PDI probe (blue and red) with gaussian distribution cut at $R = 0$ superimposed.

Detailed comments

1. p.3, l. 29 It's not clear what 'They' in the sentence 'They were smaller ...' is referring to here: 'Swiss cheese' or 'cloud holes' ?

   **It was referring to cloud holes. This part was rephrased.**

2. p.6, eq. (2) Could the authors provide a reference for this? Or at least some more motivation?

   **The reference to the original work of Burgers was provided.**

3. p.9, l. 2 What is $r_0$? Does it have any physical relevance? It would be helpful if $\varphi(x)$ were defined so that its dependence on the parameters in the problem is made clear; it has an important role in the analysis.

   **$r_0^*$ is just the notation for arbitrary solution of the Eq.6. The function $\varphi(r_0^*)$ has no parameters. It was added expicitely in the text.**

4. p.9, l.12 I didn't understand the sentence beginning 'Particle motion ...' especially 'separated from the motion 2D space'.

**It means that equations describing particle motion in the plane perpendicular to the vortex axis separate from the equation describing motion along vortex axis. This part was rewritten to make it clear.**

5. p.11, eq. (9) For the linearization of eq. (6) I obtained (). While this differs from eq. (9) for non-zero $A$, it does give me $r^* = 4\pi Sv$ in the limit $A \to 0$ in agreement with eq. (9).

**We provide this calculation in the revised manuscript.**

6. p.11, l.16 I didn't understand the first sentence '... splits in parts'. Two parts?

**This part was rephrased accordingly.**

7. p. 11, l. 23+ Why are the conclusions 'rough' ? I didn't follow all of the logic here: as the influence of gravity increases i.e. $gsin\theta$ increases or the vortex core size, $\delta$, increases (or both) then $B$ decreases. Decreasing B means decreasing Amax which is consistent with increasing $\Gamma$ (circulation). My understanding is that $A < A_{max}$ for void creation. So as either $gsin\theta$ or $\delta$, (or both) increase it becomes harder for voids to form unless the circulation increases appropriately. So I agree with the first conclusion so long as increasing minimum circulation equates to decreasing $A_{max}$ .
**We reformulated the text to make the understanding easier. Increasing minimum circulation equates to decreasing $A_{max}$ so conclusions are the same as before.**

A further consequence of decreasing $B$ is decreasing $St_1$ and $\Delta St$ since the latter is proportional to $B^{1/3}$. This implies that, for fixed $\tau_f$ (circulation), the minimum particle size decreases with decreasing $B$ as the second point suggests though it is not written very clearly. Since $\Delta St$ decreases with decreasing $B$ this suggests the range of particle size decreases with decreasing $B$ as suggested by the latter part of the third point. But decreasing $B$ corresponds to increasing $\Gamma$ which seems to contradict the first part of the third conclusion. Perhaps I have missed something in the analysis of par. 3 and 4; I would welcome more explanation.
**$\tau_f$ is not circulation (it depends on the vortex core size $\delta$ as well). We think this misunderstanding caused the confusion. Total circulation of Burgers vortex is $\Gamma = \nu/A$.**

8. p.12, l.5 The last part of this sentence doesn't read well.
**It was rewritten.**

9. p.12, §6 Which equations are the numerical simulations solving? A Burgers vortex or the Navier-Stokes equations?
**The simulations were solving particle motion equations in Burgers vortex field. It was stated explicitly in the text in section 5.**

10. p. 12, l. 32 Z and D should be defined: are they imposed on the simulation or simply typical values?

**The values were imposed on the simulations to let us account for the void creation of sizes similar to observed.**

11. p.16, l.14 My understanding of the analysis of par.3 and 4 is that $A < A_{max}$ for void creation yet here you are saying the opposite.

**The simulation set 2) in previous version was made with a mistake. The manuscript was changed in such a way that it does not present simulations with $A > A_{max}$ any more.**

**Review 4**

The paper presents field experimental observation of the so defined "cloud voids" and attempts to provide physical mechanism for this phenomena based on the simple model of inertial particle expulsion from strong, thin coherent vortices, using Burger vortex as model. The field observation of the cloud voids is in itself new and interesting and worth disclosing. However, the link to expulsion from coherent vortices is, although plausible, not as convincing as suggested in the Abstracts and Conclusion of the paper. I would recommend a somewhat weaker conclusion unless more work is done to strengthen this interpretation.

**We weakened the conclusion in the accordance to the reviewer suggestion**

In relation to this and other weaknesses, below are my detailed comments:

1. According previous works such as the cited Mouri et al (2000), coherent intense vortices are observed to have diameter not more than about 20 times the Kolmogorov microscale (as reported by Mouri et al.). This is consistent with the authors' own survey as they wrote "proportionality constant in Eq. 1 is in the range $m \in [3.5, 24]$". Their observed void size (centimeters) in this report is however about a hundred times the corresponding Kolmogorov length (see Table 1.). This calls for an explanation. E.g. are larger vortices expected in atmospheric conditions?

**We distinguish between the visible void size, dependent on effect of Mie scattering and vortex radii (Tab.3) which is close to 1cm, e.g 20 times of the Kolmogorov size in accordance to Mouri et al.**

2. As the authors stated, these coherent vortices are "severely intermittent". What could be said on the prevalence of the observed voids at the site? And could this be reconciled with known intermittency, perhaps in terms of "large-scale organization of the small-scale intermittent structures" ?

**We do not have enough documented observations to discuss this issue.**

3. Ideally, it would be very helpful, if the velocity or vorticity field around the observed void is also concurrently estimated if not measured. If that is not the case, it would be helpful to establish the presence of intense vortices, long lived enough

and having similar diameters under similar conditions, at the site.

**Unfortunately, turbulence as occurred in the experimental site cannot be reproduced in the laboratory. The authors are working on the better instruments (light-sheets and visualization techniques), reliably working in harsh experimental environment to collect more data.**

4. The Stokes number used in simulation 1) is $St = 1.45$. I was hoping to see St similar to the experimental value (similarly for $S_v$) for a better comparison (is this not the purpose, why?, in any case I recommend that). In relation to this, I don't know if it is meaningful to claim "visible void radii are rather $\approx 2 - 2.5\ cm$, which seems close to the experimental values" (line-4, pg.16) when parameters are not matched.

   **Our purpose was to prove that simulating an image of a void of diameter close to experimentally observed is possible. Calculation of $St$ and $Sv$ both use vortex characteristic times: turnover time and sedimentation time. In 3D real turbulence flow estimates of $St$ and $Sv$ are based on Kolmogorov timescale which is a global value. The single vortex in intermittent flow however can have completely different local parameter values. Comparison of this numbers was provided rather to convince the reader that they cannot use them as leading indicators of particle behaviour. We elaborate on this topic in the text. Please note that the numerical simulation examples changed in the revised manuscript (simulation set 1) is almost the same, but simulation set 2) differs from the one before).**

5. I have difficulties understanding Figure 5 and the corresponding explanation in the text. In particular, I am not sure how to interpret the dashed-lines. Better exposition is welcomed here.

   **Dashed lines present contours of periodic orbit radius solutions for a few chosen orbit radii: 0.5 cm, 1.5 cm, 3cm. Colors refer to particle size for which this contours were calculated. We reformulated the whole paragraph and we hope it is now clear.**

6. Line-15, pg 16: "Comparison of the modeled and observed voids led . . ..". observe refers to the field data or the "visible void" in the simulation?

   **It refers to the field data. It was clarified in the text.**

7. Line-4, pg.16 : "0-1.5 cm; however, . . ..". Is zero a typo here?
   **Yes, this was a typo.**

[revised manuscript text omitted]

---

## Author Response (AR2)

**Authors response to editor suggestions - minor revision**
**Turbulence Induced Cloud Voids: Observation and Interpretation**
**March 19, 2019.**

Katarzyna Karpinska[1], Jonathan F.E. Bodenschatz[2], Szymon P. Malinowski[1], Jakub L. Nowak[1], Steffen Risius[2], Tina Schmeissner[4], Raymond A. Shaw[3], Holger Siebert[4], Hengdong Xi[2], Haitao Xu[2], and Eberhard Bodenschatz[2]

[1]Institute of Geophysics, Faculty of Physics, University of Warsaw, Warsaw, Poland
[2]Max Planck Institute for Dynamics and Self-Organization, Goetingen, Germany
[3]Michigan Technological University, Houghton, Michigan, USA
[4]Leibniz Institute for Tropospheric Research, Leipzig, Germany

**Editor suggestions**

1. Referee no.1 asked for shortening of Sections 3.1 and 4.1. However, I do not have any constructive suggestions how to reduce their length. Thus, I leave it up to you whether you concur with the reviewer's suggestion.

2. However, contrary, I think that the separation into that many subsections in Sections 3 and 4 is not always very logical and useful. Very short subsections containing only one or two sentences (such as Section 3.1 or 3.2.2) should be generally avoided. Instead, I suggest combining Sections 3.1 and 3.2 using a descriptive header, e.g. Motion in and along the vortex axis (or similar). That way you would also avoid the rather (too) short headers of sections 3.2.1 and 3.2.2.

3. The readability and clarity may be also improved by replacing 'below' (as in e.g., 'results are described in detail below') by referring to the respective section number instead or by adding 'in the next paragraph/following' (or similar), respectively.

4. p. 12, l. 8/9: What is the difference between the 'first' and 'following' subsection? Or do you mean the 'first' and 'the one following the first'? Better refer to Section numbers to avoid this confusion. (Also note that the second 'subsection' in l. 9 is misspelled.)

5. Your response to Referee 2's general comment was good ("We agree with the reviewer that we use..."). However, I think these caveats and intentions should be also expressed in Section 7.

**Authors' response**

1. We have already revised sections 3 and 4 in response to another reviewer request to provide more details and interpretation. We feel that a reasonable balance between concision and explanation has been achieved.

2. We removed the subsections and added some explanations within the section to smooth the way of reasoning.

3. The manuscript was changed according to this suggestion.

4. We agree that these expressions were confusing. We revised the text respectively.

5. We incorporated this general comment into the summary in Section 7.

[revised manuscript text omitted]